# PeerJ

# Genome evolution in an ancient bacteria-ant symbiosis: parallel gene loss among *Blochmannia* spanning the origin of the ant tribe Camponotini

Laura E. Williams[1,*] and Jennifer J. Wernegreen[1,2]

[1] Duke Center for Genomic and Computational Biology, Duke University, Durham, NC, USA
[2] Nicholas School of the Environment, Duke University, Durham, NC, USA
[*] Current affiliation: Department of Biology, Providence College, Providence, RI, USA

Corresponding author
Jennifer J. Wernegreen,
j.wernegreen@duke.edu

## ABSTRACT

Stable associations between bacterial endosymbionts and insect hosts provide opportunities to explore genome evolution in the context of established mutualisms and assess the roles of selection and genetic drift across host lineages and habitats. *Blochmannia,* obligate endosymbionts of ants of the tribe Camponotini, have coevolved with their ant hosts for ∼40 MY. To investigate early events in *Blochmannia* genome evolution across this ant host tribe, we sequenced *Blochmannia* from two divergent host lineages, *Colobopsis obliquus* and *Polyrhachis turneri*, and compared them with four published genomes from *Blochmannia* of *Camponotus sensu stricto*. Reconstructed gene content of the last common ancestor (LCA) of these six *Blochmannia* genomes is reduced (690 protein coding genes), consistent with rapid gene loss soon after establishment of the symbiosis. Differential gene loss among *Blochmannia* lineages has affected cellular functions and metabolic pathways, including DNA replication and repair, vitamin biosynthesis and membrane proteins. *Blochmannia* of *P. turneri* (i.e., *B. turneri*) encodes an intact DnaA chromosomal replication initiation protein, demonstrating that loss of *dnaA* was not essential for establishment of the symbiosis. Based on gene content, *B. obliquus* and *B. turneri* are unable to provision hosts with riboflavin. Of the six sequenced *Blochmannia, B. obliquus* is the earliest diverging lineage (i.e., the sister group of other *Blochmannia* sampled) and encodes the fewest protein-coding genes and the most pseudogenes. We identified 55 genes involved in parallel gene loss, including glutamine synthetase, which may participate in nitrogen recycling. Pathways for biosynthesis of coenzyme A, terpenoids and riboflavin were lost in multiple lineages, suggesting relaxed selection on the pathway after inactivation of one component. Analysis of Illumina read datasets did not detect evidence of plasmids encoding missing functions, nor the presence of coresident symbionts other than *Wolbachia*. Although gene order is strictly conserved in four *Blochmannia* of *Camponotus sensu stricto*, comparisons with deeply divergent lineages revealed inversions in eight genomic regions, indicating ongoing recombination despite ancestral loss of *recA*. In sum, the addition of two *Blochmannia* genomes of divergent host lineages enables reconstruction of early events in evolution of this symbiosis and suggests that *Blochmannia* lineages may experience distinct, host-associated selective pressures. Understanding how

evolutionary forces shape genome reduction in this system may help to clarify forces driving gene loss in other bacteria, including intracellular pathogens.

## INTRODUCTION

The evolution of stable mutualisms between bacteria and insects has occurred many times and involves phylogenetically diverse lineages (*Moran, McCutcheon & Nakabachi, 2008*; *Moya et al., 2008*; *Kikuchi, 2009*; *Douglas, 2015*). In many of these symbioses, the limited diets of insect hosts are supplemented by an intracellular bacterial partner. Variation in dietary requirements among insect hosts likely results in differing selective pressure on endosymbiont genes, which in turn impacts endosymbiont genome evolution. For example, *Buchnera aphidicola* supplement the carbohydrate-rich plant sap diet of aphids with amino acids (*Shigenobu et al., 2000*), whereas *Wigglesworthia* species supplement the blood diet of tsetse flies with vitamins and other cofactors (*Akman et al., 2002*). By contrast, cockroaches and ants of the tribe Camponotini are generally considered omnivores with complex diets. Their bacterial partners, *Blattabacterium* and *Blochmannia*, respectively, synthesize essential amino acids and participate in nitrogen recycling (*Gil et al., 2003*; *Lopez-Sanchez et al., 2009*; *Sabree, Kambhampati & Moran, 2009*). Nitrogen recycling in *Blattabacterium* occurs via urease and glutamate dehydrogenase, whereas in *Blochmannia*, it occurs via urease and glutamine synthetase, although the gene encoding the latter enzyme is missing in some *Blochmannia* species (*Williams & Wernegreen, 2010*). In the case of *Blochmannia*, its nutritional role may be most important at particular stages of the host's lifecycle (*Zientz et al., 2006*; *Feldhaar et al., 2007*; *Stoll et al., 2010*).

Genomes of established endosymbionts like *Buchnera*, *Wigglesworthia* and *Blochmannia* are typically characterized by high AT content, elevated mutation rates and extreme stability of gene order (*Moran, McCutcheon & Nakabachi, 2008*). Recombination-related genes are often lost during endosymbiont evolution, and very few recombination events are evident in most obligate endosymbiont lineages (*Sloan & Moran, 2013*). Genome reduction in established endosymbionts occurs via degradation and loss of individual genes. Gene loss is likely shaped by both relaxed selective pressure due to the stable intracellular niche and genetic drift due to small effective population sizes and bottlenecks during vertical transmission of symbionts (*Andersson & Kurland, 1998*; *Moran, McCutcheon & Nakabachi, 2008*). The balance between these forces may shift over time for a given association, from the initial acquisition event through the ongoing evolution of the symbiosis.

To explore genome reduction and evolution in the context of a long-term endosymbiosis, we sequenced two deeply divergent lineages of Candidatus *Blochmannia*, which are obligate bacterial endosymbionts of ants of the tribe Camponotini (*Sauer et al., 2000*). *Blochmannia* are closely related to free-living Enterobacteriaceae such as *Escherichia coli*

and form a clade with other obligate endosymbionts including *Baumannia*, *Sodalis* and *Wigglesworthia* (*Herbeck, Degnan & Wernegreen, 2005*; *Husnik, Chrudimsky & Hypsa, 2011*). The presence of *Blochmannia* in multiple extant ant genera of Camponotini points to a single colonization event in the ancestral lineage as the origin of the symbiosis (*Sameshima et al., 1999*; *Wernegreen et al., 2009*). Based on phylogenetic evidence, the ancestor of *Blochmannia* may have been a facultative symbiont of insects (*Herbeck, Degnan & Wernegreen, 2005*; *Wernegreen et al., 2009*).

The ant hosts of the four previously sequenced *Blochmannia* (*Camponotus chromaiodes*, *C. floridanus*, *C. pennsylvanicus* and *C. vafer*) belong to *Camponotus sensu stricto*. Whereas these host species span the origin of *Camponotus* ∼16–20 MY, the association between *Blochmannia* and the tribe Camponotini is at least twice that old, on the order of 40 MY. To reconstruct earlier events in the evolution of this symbiosis, we sequenced the genomes of *Blochmannia* from two divergent lineages in the tribe Camponotini: the genus *Polyrhachis* and the *Colobopsis* lineage.

Though *Colobopsis* is formally considered a subgenus of *Camponotus*, phylogenetic analysis of seven nuclear gene fragments revealed it is a separate lineage from *Camponotus* (*Brady et al., 2006*; *Moreau & Bell, 2013*). These studies demonstrated that *Colobopsis* diverged early in the evolution of the tribe Camponotini, and results are consistent with *Colobopsis* being the sister group of all other Camponotini sampled, although relationships in this part of the tree were difficult to resolve conclusively. Subsequent work has further suggested that *Colobopsis* is likely the sister group of all other extant camponotines (PS Ward, pers. comm., 2014). In contrast to *Camponotus* species with published *Blochmannia* genomes, *C. obliquus* lives in small twigs and branches, often in the canopy. *Polyrhachis* diverged later than *Colobopsis* and is found in the Old World. The deep evolutionary divergence of these host lineages, as well as differences in their geographic range and habitats, provide a valuable opportunity to investigate the evolutionary trajectories of *Blochmannia* across the Camponotini and to clarify ancient events that shaped this 40 MY old ant-bacterial partnership.

## MATERIALS AND METHODS

### Preparation of genomic DNA

A single colony of *C. obliquus* was collected near Morehead City, North Carolina, USA by B Guénard, and *P. turneri* was collected near Townsville, Australia by SKA Robson (see 'Acknowledgments'). Voucher specimens were deposited in the Bohart Museum of Entomology, University of California, Davis (UCDC), corresponding to voucher ID numbers CASENT0221021 (*C. obliquus*) and CASENT0220426 (*P. turneri*). We used the Qiagen DNeasy Blood and Tissue Kit to prepare genomic DNA from a pooled sample of seven eggs, two larvae, five pupae, eight minor workers, six major workers and five female alates for *C. obliquus* and a pooled sample of three worker gasters for *P. turneri*.

## Sequencing and assembly of *Blochmannia* genomes

*C. obliquus* gDNA was sequenced on an Illumina HiSeq (Illumina, San Diego, California, USA) to generate 100 bp paired end reads. We modified filter_reads.py (https://github. com/nickloman/xbase/blob/master/short-read-assembly/filter_reads.py) for Sanger FASTQ format and used the script to remove paired reads with any bases of quality score <30, which retained 6,184,892 read pairs. We assembled this filtered read dataset using Velvet v1.2.07 (*Zerbino & Birney, 2008*) with a hash length of 61, exp_cov 200, cov_cutoff 20 and scaffolding turned off. This generated 566 contigs, one of which aligned to *B. pennsylvanicus* using MAUVE (*Darling, Mau & Perna, 2010*). We observed an overlap of 116 bp between the contig ends, which suggested that *de novo* assembly had produced a closed genome. To test this, we used Mosaik (*Lee et al., 2014*) to align the filtered read dataset against a 1,260 bp sequence encompassing the joined contig ends and flanking regions. This alignment produced no zero coverage regions, which confirmed that the single contig was the closed *B. obliquus* genome.

To finish the genome, we used a two-step process applying different alignment programs to confirm the majority genotype, or the base represented by the majority of reads at each position. In the first step, we used Mosaik, which allows the user to set the stringency of mismatch tolerance. We aligned the filtered read dataset against the closed genome sequence with a maximum mismatch threshold of 12 and then removed duplicate read pairs with the DupSnoop module. Using Consed (*Gordon, Abajian & Green, 1998*), we generated a questionable consensus bases report and a highly discrepant indels report for the resulting alignment, which did not identify any positions that needed editing.

The second step in our finishing process invokes the IndelRealigner module of the Genome Analysis Toolkit (GATK) to ensure accurate identification of indels (*DePristo et al., 2011*). We aligned the filtered read dataset against the closed genome sequence using BWA (*Li & Durbin, 2009*) and then processed the resulting alignment with RealignerTar-getCreator and IndelRealigner from the GATK package. We removed duplicate reads using Picard MarkDuplicates (http://broadinstitute.github.io/picard). Finally, we analyzed the processed alignment using VarScan (*Koboldt et al., 2009*) to identify positions at which the base in the closed genome sequence differed from the majority of aligned reads. We did not identify any such positions, confirming that the single contig was the closed, finished *B. obliquus* genome. Genome coverage for this alignment averaged 558x. This sequence is deposited in GenBank as accession number CP010049. The Illumina read dataset is deposited in the NCBI Short Read Archive (SRA) as SRP050154.

*P. turneri* gDNA was sequenced on an Illumina Genome Analyzer II (GAIIx; Illumina, San Diego, California, USA) to generate 150 bp paired end reads. We used DynamicTrim.pl and LengthSort.pl from the SolexaQA package (*Cox, Peterson & Biggs, 2010*) to generate trimmed reads of at least 80 bp with quality score >30 for each base. The resulting trimmed read dataset included 9,590,066 read pairs. We assembled this filtered read dataset using Velvet with a hash length of 41, exp_ cov 200, cov_cutoff 20 and scaffolding turned off. This generated 8,275 contigs, four of which aligned to *B. pennsylvanicus* using MAUVE. To close

the four gaps, we generated Sanger sequencing reads and used Phred/Phrap/Consed to assemble and manually examine the sequence.

To finish the *B. turneri* genome, we followed the two-step process described above for *B. obliquus*. In the first step, we corrected nine positions based on alignment of the full read dataset against the closed genome sequence using Mosaik. In the second step, we aligned the full read dataset against the corrected sequence using BWA followed by processing with GATK. We analyzed the processed alignment with VarScan and corrected a single position. Genome coverage for this alignment averaged 1223x. The closed, finished genome sequence is deposited in GenBank as accession number CP010048. The Illumina read dataset is deposited in the NCBI Short Read Archive (SRA) as SRP050161.

## Annotation of *Blochmannia* genomes

We used an annotation engine hosted by the Institute for Genome Sciences (IGS) at the University of Maryland School of Medicine to generate an automated annotation of each genome sequence (*Galens et al., 2011*), which we then manually curated within the MANATEE framework (http://manatee.sourceforge.net/igs/). Protein-coding genes predicted by the annotation engine were removed if they lacked a Blast-Extend-Repraze (BER) alignment score $<10^{-5}$ to a protein-coding gene from outside of *Blochmannia*. We manually examined possible frameshifted genes flagged by the annotation engine. For genes with frameshifts in homopolymer tracts, we included the likely position of the frameshift in the GenBank annotation. Because the frameshifts may be corrected by polymerase slippage (*Tamas et al., 2008*; *Wernegreen, Kauppinen & Degnan, 2010*), we consider these to be intact genes.

We curated start sites using BER alignments to *Blochmannia* and closely related species. When possible, we used the gene name and symbol listed in SwissProt for the homologous gene in *E. coli* to maintain consistency with existing proteobacterial annotations. For conserved hypothetical proteins or proteins with similarity to a protein family but not a specific family member, we did not assign a gene name and refer to them using the locus tag (for example, BTURN675_020).

After curating the annotations, we analyzed intergenic regions in each genome with RFAM (*Burge et al., 2013*) and BLASTX (*Altschul et al., 1990*) to identify uncalled genes and pseudogenes. In both *B. obliquus* and *B. turneri*, RFAM identified three RNA-coding genes (*ffs*, *rnpB* and tmRNA). To identify protein-coding genes, we aligned intergenic regions to the GenBank non-redundant database using BLASTX with the low complexity filter turned off. We manually examined hits with e-value $<10^{-5}$. Pseudogenes were identified by multiple nonsense mutations, frameshifts and/or gaps. Pseudogenes that aligned to intact homologs with >60% coverage had at least two nonsense mutations. We annotated pseudogene coordinates using the boundaries of the BLASTX alignments. Analysis of *B. obliquus* intergenic regions detected two genes (*cyoD* and *sdhD*) and 15 pseudogenes (*dnaA*, *engD*, *glnA*, *pdxA*, *pdxB*, *pdxJ*, *ribA*, *ribB*, *ribC*, *ribD*, *secD*, *secF*, *topA*, *yigB* and *uvrD*). Analysis of *B. turneri* intergenic regions detected five genes (*cyoD*, *infA*, *rpmJ*, *ycaR*

and *yidD*), one frameshifted gene (*ybeY*) and one pseudogene of a hypothetical protein (BTURN675_514).

## Phylogenetic analysis

We classified genes into six MultiFun categories (cell processes, cell structure, information transfer, metabolism, regulation, and transport) by searching for the gene name in the EcoCyc database. Some genes are assigned to more than one MultiFun category, and we included all categories listed for each gene. If the gene had no associated MultiFun terms in EcoCyc or if it had no gene name, such as BPEN_040, we considered the gene unclassified.

To construct a phylogeny, we chose *Baumannia* (NC_007984), *Hamiltonella* (NC_012751) and *Sodalis* (NC_007712) as outgroups. We identified orthologs of *Blochmannia* genes in these genomes using the Reciprocal Smallest Distance (RSD) algorithm (*Wall, Fraser & Hirsh, 2003*) with default values for divergence (0.8) and e-value ($10^{-5}$). For each of the six MultiFun categories, we randomly selected five genes present in all taxa. There were no duplicates in the resulting set of 30 genes. We excluded *B. chromaiodes* from the phylogeny because its genome sequence is 98.0% identical to that of *B. pennsylvanicus* (*Williams & Wernegreen, 2013*). We used TranslatorX (*Abascal, Zardoya & Telford, 2010*) and MAFFT (*Katoh et al., 2005*) to construct a multiple sequence alignment for each gene, which we then trimmed with ZORRO (*Wu, Chatterji & Eisen, 2012*). We concatenated the trimmed amino acid alignments and used MrBayes v3.2.1 (*Ronquist & Huelsenbeck, 2003*) to construct a majority rule consensus tree. To test whether maximum likelihood methods produced the same topology, we analyzed the same dataset using MEGA v6.0 (*Tamura et al., 2013*) with a cpREV + G + I + F amino acid substitution model, which had the lowest Bayesian Information Criterion (BIC) score, and assessed branch support using 500 bootstrap replicates.

## GC skew and DnaA box motif search

For *B. pennsylvanicus* (NC_007292), *B. obliquus,* and *B. turneri*, we used DNAPlotter (*Carver et al., 2009*) to construct plots of GC skew with 500 bp window size and 50 bp step size. We used Pattern Locator (*Mrazek & Xie, 2006*) to search these three genome sequences for the consensus DnaA box motif TTWTNCACA.

## BLAST analysis

Because we prepared genomic DNA from whole ants or gasters, the Illumina read datasets include coverage of genomes other than *Blochmannia*, such as the ant host nuclear and mitochondrial genomes. To determine if the *de novo* assemblies included contigs from potential *Blochmannia* plasmids or other bacterial symbionts, we used BLASTN to align all ≥500 bp contigs to the GenBank non-redundant database, limited to bacteria (taxid 2). For contigs with at least one hit of >30% coverage and an evalue of $<10^{-5}$, we aligned each contig against the full non-redundant database and examined the top hits.

To test whether the Illumina read datasets contain evidence of riboflavin biosynthesis genes, we constructed BLAST databases of the *C. obliquus* and *P. turneri* read datasets. For *C. obliquus*, we built a BLAST database using the unaligned reads file generated

**Table 1** Genome statistics of six sequenced *Blochmannia*.

| Genome | Size | GC content (%) | Genes (inc. pseudo) | Protein coding | tRNA | rRNA | Other RNA | Pseudogenes | Frameshifted genes |
|--------|------|----------------|---------------------|----------------|------|------|-----------|-------------|--------------------|
| *B. obliquus* | 773,940 | 27.4 | 642 | 584 | 37 | 3 | 3 | 15 | 4 |
| *B. turneri* | 749,321 | 29.1 | 634 | 589 | 38 | 3 | 3 | 1 | 7 |
| *B. chromaiodes* | 791,219 | 29.5 | 658 | 609 | 40 | 3 | 3 | 3 | 4 |
| *B. pennsylvanicus* | 791,654 | 29.6 | 658 | 609 | 40 | 3 | 3 | 3 | 4 |
| *B. floridanus* | 705,557 | 27.4 | 637 | 590 | 37 | 3 | 3 | 4 | 4 |
| *B. vafer* | 722,585 | 27.5 | 631 | 587 | 37 | 3 | 2 | 2 | 8 |

by Mosaik during the initial finishing step (see above), thereby excluding most reads originating from the *rib* pseudogenes within the *B. obliquus* genome. We compiled a set of query genes representing orthologs of conspicuously absent genes, including *ribABDEF* from *B. pennsylvanicus* (NC_007292), *Baumannia* (NC_007984) and *E. coli* MG1655 (NC_000913), *ribH* from *B. pennsylvanicus* and *Baumannia, ribC* from *E. coli* and the *Camponotus floridanus* gene for EF-1alpha-F2 (EFN72500). For analysis of *C. obliquus* reads, we also included *ribF* and *rib* pseudogenes from *B. obliquus* in the query set. We used BLASTN to align the query genes against both BLAST databases and examined the output for alignments with e-values $<10^{-5}$.

Certain gene distribution patterns may be explained by either parallel gene loss or acquisition via horizontal gene transfer. To test the hypothesis of horizontal gene transfer, we aligned each of the 55 genes with such patterns against the GenBank non-redundant database using BLAST. We first used BLASTN, and if this search did not return significant hits, we used BLASTX. We examined the taxonomic assignments of the top hits to identify genes with high scoring alignments to bacteria outside of the Enterobacteriaceae, which may support the hypothesis of horizontal gene transfer.

## RESULTS

### *B. obliquus* has the fewest protein-coding genes and most pseudogenes

The size and GC content of the *B. obliquus* and *B. turneri* genomes are within the ranges observed for *Blochmannia* of *Camponotus sensu stricto* (Table 1). Although *B. obliquus* is on the upper end of the size range, it has the fewest protein-coding genes of the six sequenced *Blochmannia*. We also detected 15 pseudogenes in *B. obliquus*, which is an unusually high number for *Blochmannia* (Table S1). In both *B. obliquus* and *B. turneri*, we identified genes that have frameshifts in homopolymer tracts but otherwise are expected to encode a full-length protein. We consider these genes to be functional, because they may be corrected by polymerase slippage during transcription and expressed as full-length proteins (*Tamas et al., 2008*; *Wernegreen, Kauppinen & Degnan, 2010*).

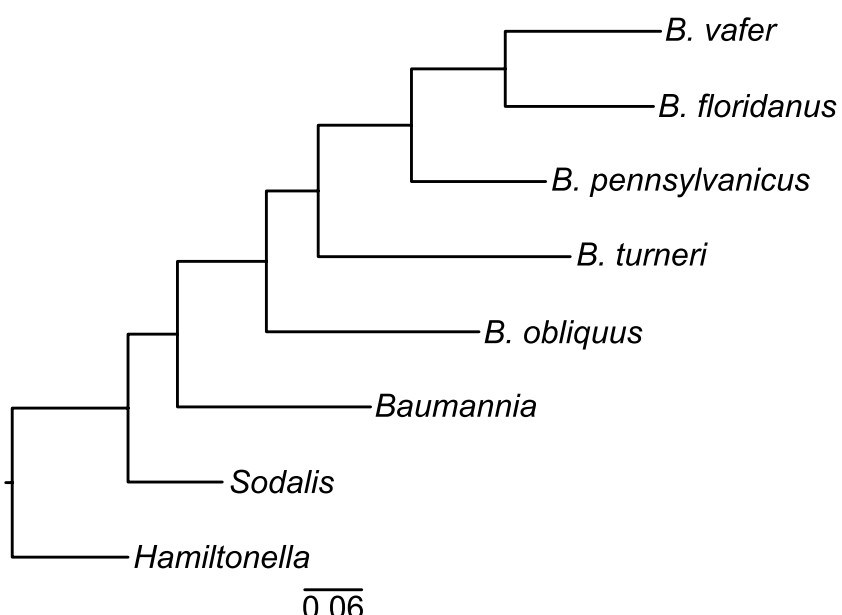

**Figure 1** ***Blochmannia* phylogeny.** The phylogeny was constructed by Bayesian analysis of concatenated amino acid sequence alignments for 30 genes. The phylogeny is artificially rooted on the branch leading to *Hamiltonella*. All nodes have 100% posterior probability. The scale bar shows amino acid substitutions.

### *B. obliquus* is the earliest diverging *Blochmannia* lineage sequenced

To determine the evolutionary relationships of the *Blochmannia* lineages, we constructed a phylogeny of 30 randomly selected protein-coding genes shared among *Blochmannia* genomes and the outgroups *Baumannia*, *Hamiltonella* and *Sodalis*. Using Bayesian methods, the resulting phylogeny has 100% posterior probability at all nodes (Fig. 1). Maximum likelihood methods produced the same topology with ≥99% bootstrap values. In this phylogeny, *B. turneri* and *Blochmannia* of *Camponotus sensu stricto* are more closely related to each other than to *B. obliquus*. This topology is congruent with recent phylogenies of the ant host taxa that show *Colobopsis* as a separate lineage rather than a subgenus of *Camponotus* (*Brady et al., 2006*). Hereafter, we use the phrase "*Blochmannia* of *Camponotus*" to mean *Blochmannia* of *Camponotus sensu stricto*, which does not include *Colobopsis*.

### Assemblies show evidence of *Wolbachia* but no *Blochmannia* plasmids or other bacterial symbionts

Because we prepared genomic DNA from whole ants or gasters, the Illumina read datasets include coverage of genomes other than *Blochmannia*. To determine whether the assemblies generated by Velvet included contigs from other symbionts or possible plasmids, we aligned all contigs ≥500 bp against the GenBank non-redundant database using BLASTN. We limited our first search to bacteria (taxid 2). For contigs with hits of evalue $<10^{-5}$ and >30% query coverage, we aligned them against the full database. The top hits from this search are shown in Table S2.

Analysis of the *C. obliquus* assembly identified 15 contigs with BLAST hits that met the criteria outlined above. Three of these are from the ant nuclear genome and one is from the ant mitochondrial genome (Table S2). The best-scoring hits for the remaining 11 contigs are all matches to *Wolbachia*, an endosymbiont found in many insect species (*Werren, Baldo & Clark, 2008*), including ants (*Russell et al., 2012*). We did not detect evidence of any other symbiont or any *Blochmannia* plasmids.

Analysis of the *P. turneri* assembly identified eight contigs with BLAST hits that met the criteria outlined above. Four of these are from the ant nuclear genome and three are from the ant mitochondrial genome (Table S2). The best-scoring hit for the remaining contig is to *B. pennsylvanicus*. To examine this further, we aligned the contig to the *B. turneri* genome sequence, which produced a much better alignment with 100% coverage and 85% identity. The contig spans the 3′ end of *prfB*, a 19 bp intergenic region and the 5′ end of *lysS*. Coverage of this contig averages 54x, whereas coverage of the same region in the *B. turneri* genome sequence averages 1,380x. We aligned the full read dataset against both the *B. turneri* genome sequence and the contig and then did a *de novo* assembly using only the mapped reads. The resulting assembly did not reconstruct the contig, suggesting that it was an artifact of assembly rather than evidence of a plasmid or other symbiont.

## Gene content of reconstructed Last Common Ancestor is highly conserved in divergent *Blochmannia* lineages

Using the six sequenced *Blochmannia* genomes, we reconstructed the likely gene content of the Last Common Ancestor (LCA) of these lineages. Horizontal gene transfer (HGT) is considered unlikely in *Blochmannia* due to the isolated intracellular niche of these bacteria. Conservation of gene order in *Blochmannia* genomes supports the hypothesis that HGT is very rare. Additionally, we used BLAST to identify the best hits for genes with distribution patterns consistent with either independent loss in multiple lineages or acquisition via HGT, and we did not find evidence of HGT (see below). For these reasons, we expect very little, if any, gain of genes in *Blochmannia* lineages, and our reconstruction of the LCA includes any intact gene found in at least one of the sequenced genomes.

By this definition, the genome of the LCA consists of 690 genes (Fig. 2). Most of these genes are retained in the *Blochmannia* genomes, with 568 genes, or 82% of the gene content of the LCA, found in all six species. The complete ortholog table can be found as Table S3. Our reconstruction may underestimate LCA gene content, because it is possible that genes were lost independently from all lineages at some point after divergence from the LCA. For example, *B. chromaiodes*, *B. pennsylvanicus* and *B. obliquus* encode a pseudogene of *uvrD*, but no sequenced *Blochmannia* has an intact *uvrD*. This is the only *Blochmannia* gene found solely as a pseudogene. This gene may have been intact and functional in the LCA, but it is not represented in our reconstruction here. Sequencing of additional taxa, including *Blochmannia* from other deeply divergent ant host genera such as *Opisthopsis*, will further refine reconstruction of the ancestral lineage.

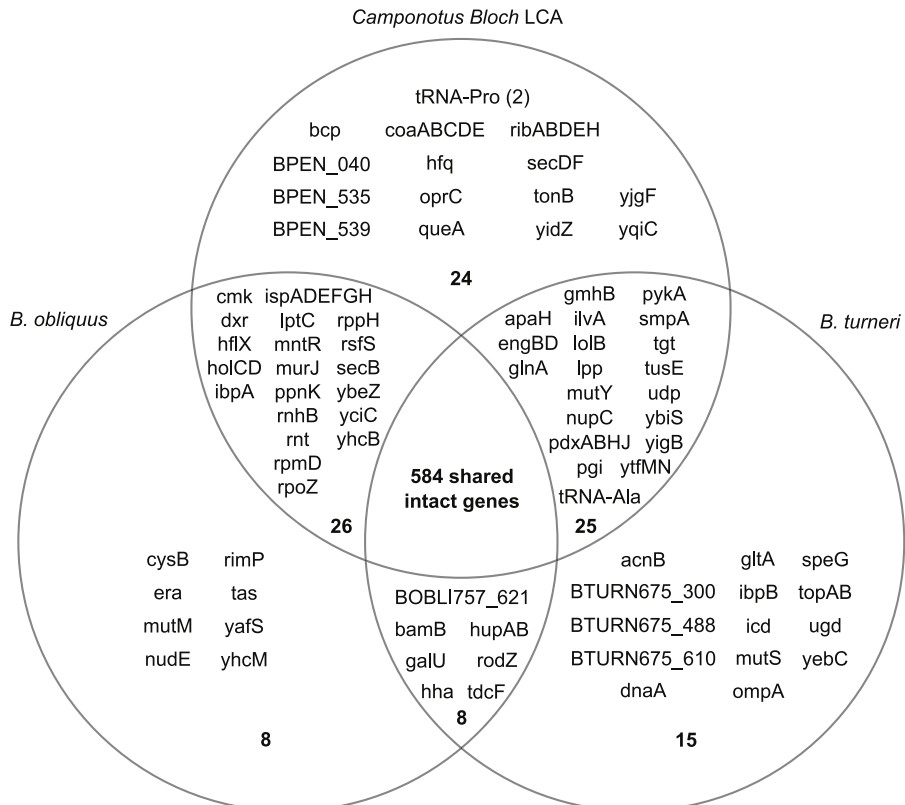

**Figure 2 Gene content of three divergent *Blochmannia* lineages.** The total number of intact genes, including genes that have frameshifts in homopolymer tracts, is shown in bold in each section. *Camponotus Bloch LCA* includes any intact gene found in at least one of the four sequenced *Blochmannia* of *Camponotus sensu stricto*. Note that *coaBC* is one gene. Although *yidC* and *yidD* are separate genes in *B. turneri* and *B. floridanus*, these two genes are fused in the other sequenced genomes and therefore counted as one gene here. Similarly, BOBLI757_064 and BOBLI757_065 are counted as one gene because they encode the two domains of bifunctional protein *hldE*.

## Gene content differences among divergent *Blochmannia* lineages include important cellular functions

### *DNA replication and repair*

The *dnaA* chromosomal replication initiation protein is intact in *B. turneri*, detectable as a pseudogene in *B. obliquus* and missing in all four sequenced *Blochmannia* of *Camponotus* (Fig. 2). In other gamma-proteobacteria, DnaA initiates replication by binding to 9-bp sequences called DnaA boxes within the origin of replication (*Zakrzewska-Czerwinska et al., 2007*). GC skew analysis predicts that the origin of replication is adjacent to *mnmG* in most *Blochmannia* species (Fig. 3), so we searched the intergenic regions flanking *mnmG* in *B. obliquus*, *B. pennsylvanicus* and *B. turneri* for the consensus DnaA box motif TTWTNCACA (*Schaper & Messer, 1995*).

We found two matches in *B. turneri*, none in *B. obliquus* and one match in *B. pennsylvanicus*. By comparison, *Buchnera aphidicola*, which retained *dnaA*, has two DnaA boxes in *oriC* (*Mackiewicz et al., 2004*); therefore, it is possible that DnaA can initiate replication in

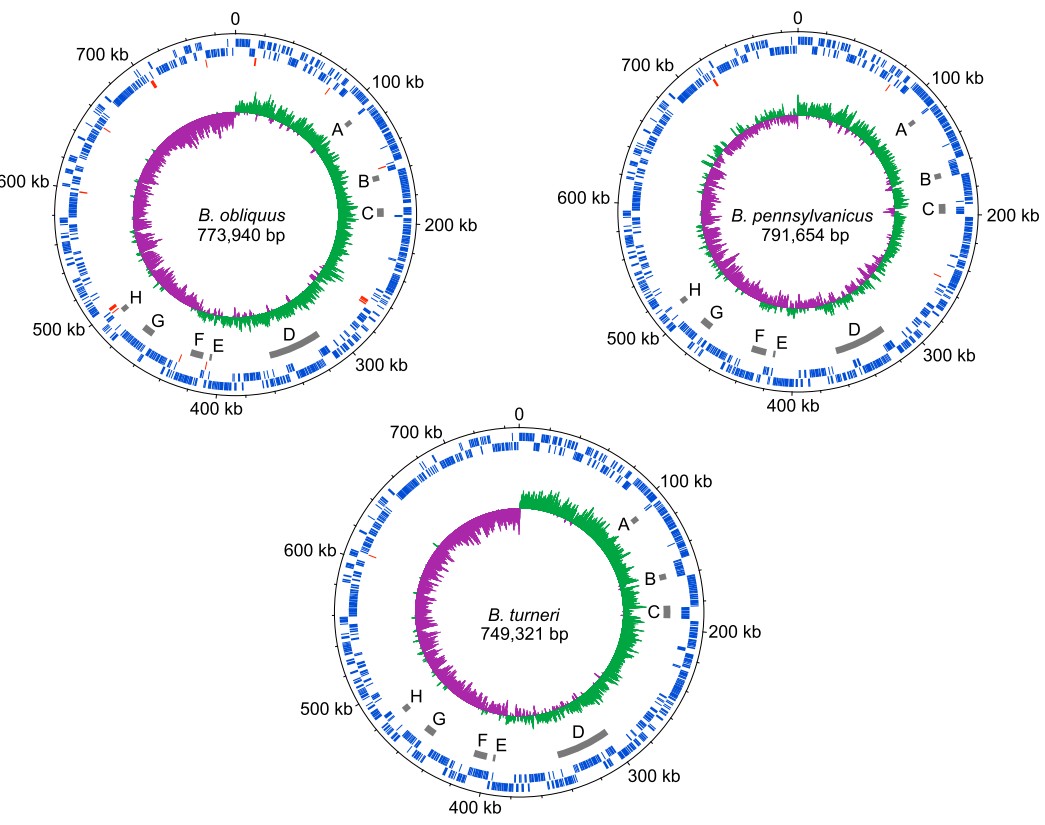

**Figure 3** *Blochmannia* **genome plots.** Plots of *B. obliquus* and *B. turneri* (sequenced in this study) and *B. pennsylvanicus* (as a representative of *Blochmannia* of *Camponotus sensu stricto*) were constructed with DNAPlotter. Position zero is set to the ATG start of *mnmG* for each genome. Major and minor tick marks on the outer circle show 100 kbp and 20 kbp increments, respectively. Tracks 1 and 2 show CDS in blue on the forward and reverse strands, respectively. Track 3 shows pseudogenes in red. Track 4 shows the eight genomic regions that experienced inversions in at least one of the lineages in grey. These regions are labeled A–H for consistency with Fig. 6. Track 5 shows GC skew calculated using 500 bp window size and 50 bp step size. Green shading above the line indicates GC skew greater than the genome average, whereas purple shading below the line indicates GC skew smaller than the genome average.

*B. turneri* with only two consensus DnaA boxes. However, the presence of a DnaA box in *B. pennsylvanicus*, which lacks *dnaA*, suggests that these motifs are not necessarily associated with DnaA function in *Blochmannia*. In fact, when we searched the entire length of each genome, we found multiple matches to the consensus DnaA box motif, which are unlikely to be involved in DnaA function as described for gamma-proteobacteria such as *E. coli* (*Hansen et al., 2006*). More work is needed to understand how the differential distribution of DnaA in *Blochmannia* species affects control of DNA replication and whether DnaA boxes play a role in replication initiation in *B. turneri*.

Loss of DNA repair mechanisms is a common characteristic of obligate intracellular symbionts of insects. In the six sequenced *Blochmannia*, two genes involved in base excision repair, *mutM* and *mutY*, are differentially distributed. *B. obliquus* encodes *mutM* but is missing *mutY*, whereas *B. turneri* and the four *Blochmannia* of *Camponotus* are missing *mutM* and encode *mutY* (Fig. 2). Both protein products act on 8-oxoG, which

can mispair with adenine (*Michaels et al., 1992*). MutM excises 8-oxoG when it is paired with cytosine, thereby initiating base excision repair. If 8-oxoG is not removed prior to replication, MutY removes the mispaired adenine, enabling repair. Inactivation of either gene leads to an increase in GC-to-TA transversions, which may contribute to the reduced genomic GC content observed in *Blochmannia* and other insect endosymbionts (*Lind & Andersson, 2008*).

### *Vitamin biosynthesis*

Pyridoxal 5′-phosphate (PLP), the catalytically active form of vitamin B6, is an important enzyme cofactor. With very few exceptions, insects and other animals cannot synthesize vitamin B6 (*Tanaka, Tateno & Gojobori, 2005*). Biosynthesis of PLP in *E. coli* and other gamma-proteobacteria occurs via a pathway encoded by seven genes (*Mukherjee et al., 2011*). This pathway is conserved in *B. turneri* and *Blochmannia* of *Camponotus*, with the exception of *epd/gapB*, which is missing from all six sequenced *Blochmannia* (Fig. 4). The function of *epd* may be fulfilled by *gapA*, which is present in all six species and can compensate *epd* mutants in *E. coli* (*Yang et al., 1998*). In *B. obliquus*, only *dxs* and *serC* are conserved, whereas *pdxABHJ* are missing. We detected pseudogenes of three of these genes. Based on these gene losses, *B. obliquus* appears unable to provision its ant host with vitamin B6.

Riboflavin (vitamin B2) biosynthesis is another vitamin synthesis pathway with differential gene distribution in these *Blochmannia* lineages. Riboflavin is essential for synthesis of the cofactors flavin mononucleotide (FMN) and flavin adenine dinucleotide (FAD) (*Abbas & Sibirny, 2011*). *Blochmannia* of *Camponotus* encode five riboflavin biosynthesis genes (*ribABCDH*), which comprise the complete pathway in *E. coli*. By contrast, these genes are missing in both *B. obliquus* and *B. turneri*. We detected pseudogenes of *ribABCD*, but not *ribH*, in *B. obliquus*. The *ribF* gene, which encodes an enzyme that synthesizes FMN and FAD from riboflavin, is conserved in all six sequenced *Blochmannia* species.

It is possible that riboflavin biosynthesis genes are encoded on a plasmid, by a secondary symbiont (*Lamelas et al., 2011*), or even within the ant host nuclear genome (*Husnik et al., 2013*). As discussed above, we detected *Wolbachia* but no plasmids or other symbionts in the assemblies. To investigate possible alternative sources of riboflavin, we analyzed the Illumina read datasets. Using BLASTN, we aligned a query set of *rib* genes from *B. pennsylvanicus*, *Baumannia* and *E. coli* against the reads. We also included the elongation factor alpha F2 (EF-1alpha-F2) gene from *Camponotus floridanus* in the query set to assess representation of the ant host nuclear genome in the read datasets. Because *B. obliquus* has pseudogenes of four riboflavin biosynthesis genes, we used the subset of reads that did not align to the *B. obliquus* genome to build the *C. obliquus* BLAST database.

We found no evidence of intact riboflavin biosynthesis genes in the read datasets. For reads from *P. turneri* gasters, we detected no BLASTN hits with e-values $<10^{-5}$ to the query riboflavin biosynthesis genes. For reads from *C. obliquus* ants, some reads aligned to the *B. obliquus* pseudogenes. Based on examination of the BLAST alignments, these reads likely originated from the pseudogenes, but they have more mismatches than permitted by Mosaik and were included in the unaligned reads file. When we considered only reads

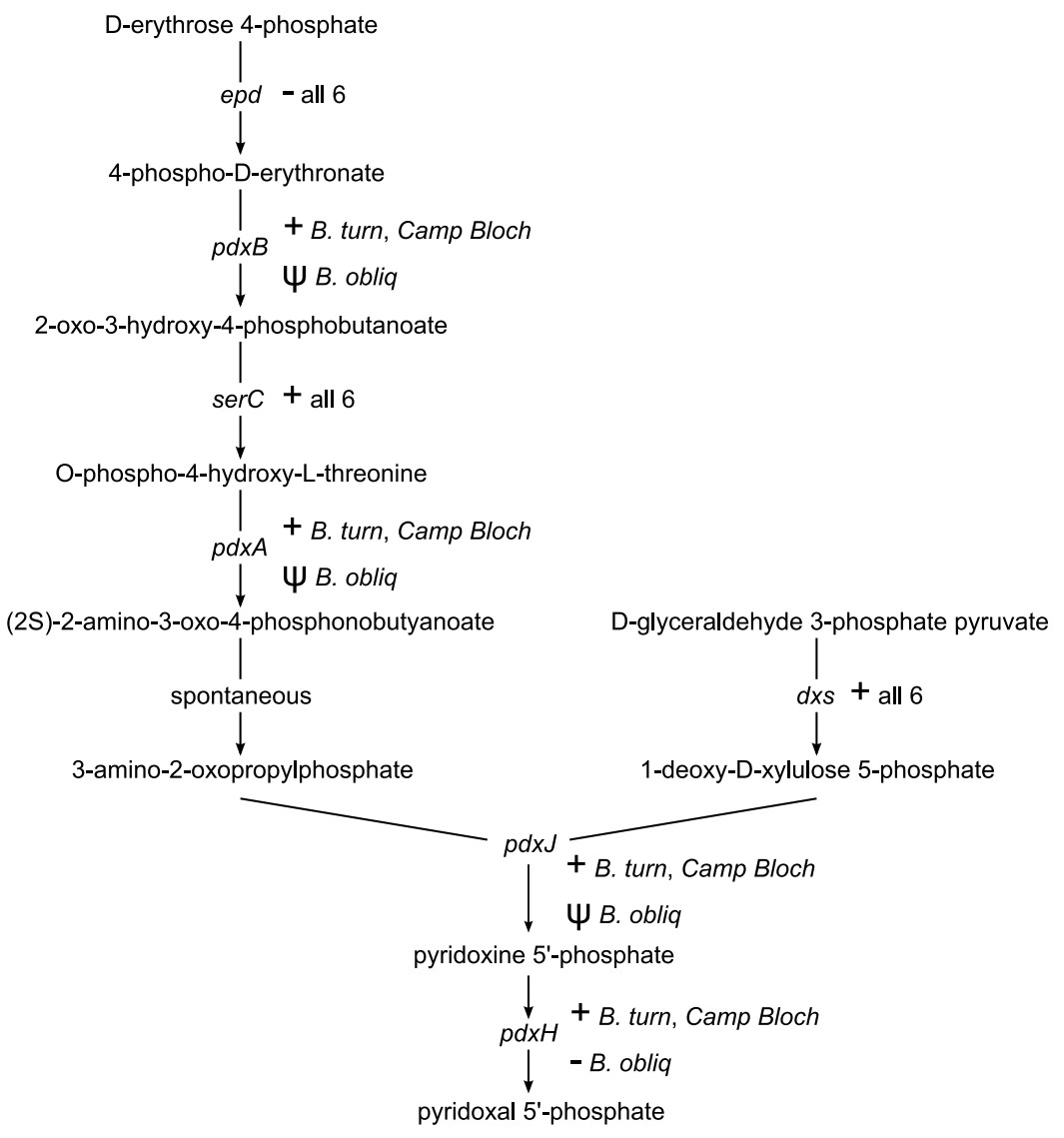

**Figure 4 Vitamin B6 synthesis encoded by six *Blochmannia* genomes.** The pathway for synthesis of vitamin B6 is annotated with gene distribution in the six sequenced *Blochmannia*. *Camp Bloch* refers to all four sequenced *Blochmannia* of *Camponotus sensu stricto*, which have the same gene content for this pathway. A plus sign indicates an intact gene, a minus sign indicates a missing gene, and a psi symbol indicates a pseudogene.

that did not align by BLASTN to the *rib* pseudogenes, we detected only three reads aligning to the query riboflavin biosynthesis genes with evalues $<10^{-5}$. A single read, but not its mate, aligned to *ribE* from *B. pennsylvanicus* with only 58% coverage and an evalue of $\sim10^{-6}$, and a single read pair aligned to *ribA* from *E. coli* with 100% identity and 100% coverage, which may be due to contamination or presence of gut-associated bacteria. These single reads contrast with the 558x coverage of the *B. obliquus* genome. For both read datasets, multiple reads aligned to the *C. floridanus* EF-1alpha gene with evalues $<10^{-20}$, confirming that the reads include coverage of the host nuclear genome. Our analysis of the

**Table 2** Tandem repeats within *tolA* of *Blochmannia* and *E. coli.*

| Genome | Length of *tolA* (bp) | Length of consensus repeat pattern (bp) | # of repeat copies | Alignment score | Coordinates of repeat region |
|---|---|---|---|---|---|
| B. obliquus | 771 | 17 | 2.1 | 52 | 81–114 |
| B. turneri | 990 | 81 | 4.4 | 699 | 205–558 |
| B. chromaiodes | 1,185 | 87 | 6.0 | 1,030 | 199–722 |
| B. pennsylvanicus | 1,185 | 87 | 6.0 | 1,046 | 200–722 |
| B. floridanus | 1,305 | 99 | 6.5 | 1,260 | 215–853 |
| B. vafer | 1,173 | 102 | 5.1 | 1,038 | 208–726 |
| E. coli MG1655[a] | 1,266 | 79 | 2.0 | 239 | 671–829 |

**Notes.**

Data from Tandem Repeats Finder (*Benson, 1999*).

[a] Four repeat patterns were identified. The top-scoring pattern is reported here.

Illumina read datasets did not reveal how these symbiotic systems compensate for the loss of riboflavin biosynthesis in the *Blochmannia* partners.

### Membrane proteins

All six sequenced *Blochmannia* encode *tolA*, an inner membrane protein and a component of the Tol-Pal system; however, there are structural differences among the species. In *E. coli*, TolA has three domains: a transmembrane domain that anchors the protein in the cytoplasmic membrane, a central periplasmic domain with high alpha-helix structure and a globular periplasmic domain (*Godlewska et al., 2009*). In *Blochmannia*, five of the six species have a stretch of 80–100 bp tandem repeats in *tolA*, which varies in length and repeat sequence among species (Table 2). By contrast, *B. obliquus* shows no evidence of this repeat region, which is also not found in the *E. coli* homolog. *E. coli tolA* has a repeat region, but it occurs in a distinct location in the protein and consists of shorter repeats (*Zhou et al., 2012*). In addition, the length of *B. obliquus tolA* (771 bp) is shorter than *tolA* in the other *Blochmannia* species (990–1,305 bp) and *E. coli* (1,266 bp).

The function of *tolA* in an insect endosymbiont such as *Blochmannia* is unknown. In *E. coli*, the Tol-Pal system is thought to interact with phage particles and colicins (*Godlewska et al., 2009*). Some structural features described for *E. coli* TolA are conserved in *Blochmannia*, such as the *N*-terminal transmembrane domain. Both the TMHMM server v 2.0 and Phobius predicted one transmembrane helix within the first 50 amino acids of TolA from each sequenced *Blochmannia* genome. Six of the seven genes comprising the two Tol-Pal operons (*ybgC-tolQ-tolR-tolA* and *tolB-pal-ybgF*) in *E. coli* are conserved in all six sequenced *Blochmannia*, with only *ybgC* missing. The Tol-Pal system may be important in *Blochmannia* for host-endosymbiont interactions.

### Phylogenetic framework reveals parallel gene losses

We placed gene losses in a phylogenetic context to identify parallel losses (Fig. 5), which we define as independent loss of the same gene in multiple lineages separated by a lineage that retained the gene. By this definition, we identified 55 genes that were subject to parallel loss in these *Blochmannia* lineages. An alternative explanation for the distribution patterns

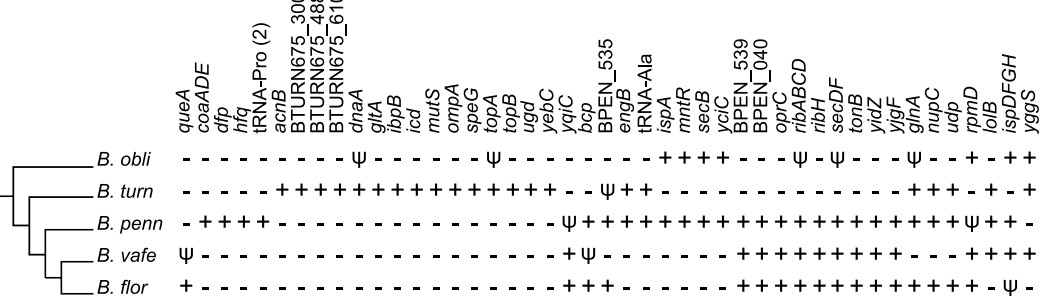

**Figure 5  Parallel gene losses.** The distribution of genes lost in multiple independent events (i.e., parallel gene losses) are shown in a phylogenetic context. A plus sign indicates an intact gene, which includes frameshifted genes, a psi symbol indicates a pseudogene and a minus sign indicates that the gene is absent. The gene content of *B. chromaiodes* and *B. pennsylvanicus* is identical; therefore, only *B. pennsylvanicus* is shown here. Acquisition via horizontal gene transfer could, in principle, explain some observed patterns; however, such transfer is unlikely in *Blochmannia*, and we found no evidence for HGT of genes listed here (see text).

of these 55 genes is acquisition by horizontal gene transfer. To test this explanation, we aligned each of the 55 genes to the GenBank non-redundant database using BLAST and identified only two genes (*icd* and *yqiC*) for which the top hits were to bacteria outside of the Enterobacteriaceae. The top BLASTN hit for *icd* was to Candidatus *Profftella*, a beta-proteobacterial endosymbiont of the Asian citrus psyllid. BLASTN did not return any significant hits for *yqiC*, but the top BLASTX hit was to *Vibrio litoralis*. Although it is possible that these genes were acquired by *Blochmannia* via horizontal gene transfer, for the purposes of our analysis we consider parallel gene loss a more likely explanation for their distribution patterns.

Because of the branching order in the *Blochmannia* phylogeny (Fig. 1), the 15 genes unique to *B. turneri* must have been independently lost in both *B. obliquus* and *Blochmannia* of *Camponotus* (Fig. 5), assuming that gene content differences reflect gene loss rather than HGT (see above). Four of these 15 genes are involved in DNA replication and repair, including the chromosomal replication initiation protein *dnaA,* the DNA mismatch repair protein *mutS* and DNA topoisomerases *topA* and *topB*. Three of the 15 genes (*acnB*, *gltA* and *icd*) encode enzymes of the tricarboxylic acid (TCA) cycle and complete this pathway in *B. turneri*. By contrast, *B. obliquus* and *Blochmannia* of *Camponotus* retained only TCA genes involved in energy generation. In general, parallel loss of these 15 unique *B. turneri* genes in *B. obliquus* and *Blochmannia* of *Camponotus* may indicate differing selective pressures on *B. turneri* and its ant host.

Multiple genes involved in metabolic pathways were subject to parallel loss in the six sequenced *Blochmannia*. Glutamine synthetase was independently lost in both *B. vafer* and *B. obliquus*. In other *Blochmannia*, this enzyme may play an important role in nitrogen recycling for the ant host (*Feldhaar et al., 2007*). Parallel loss of *glnA* suggests that this enzyme is not essential to the symbiosis and its function can be fulfilled by an alternative pathway, such as arginine biosynthesis via carbamoyl phosphate synthase (*Williams & Wernegreen, 2010*). A few parallel loss events involved genes from the same metabolic
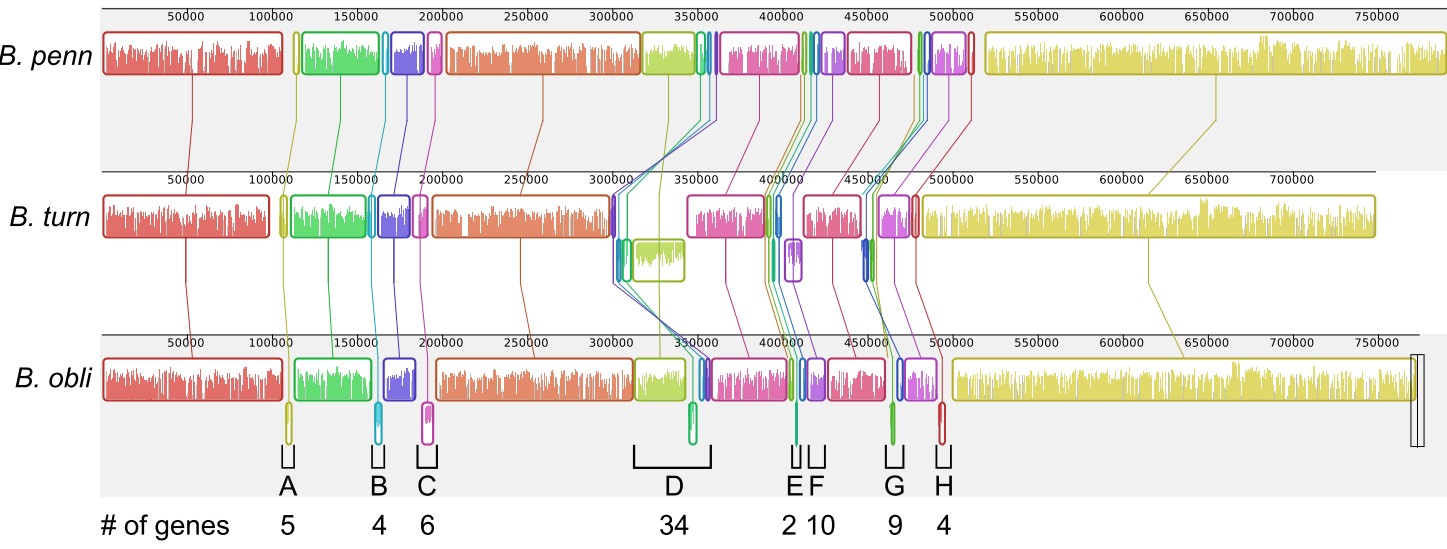

**Figure 6 Inversions in divergent *Blochmannia* lineages.** Whole genome alignment of *B. pennsylvanicus* (as a representative of *Blochmannia* of *Camponotus sensu stricto*), *B. turneri* and *B. obliquus* with progressive MAUVE shows inversions as colored blocks below the midline for each genome. The eight genomic regions that experienced inversions are labeled A–H with the number of genes involved in each also shown.

pathway, such as *coaADE* and *dfp* (coenzyme A biosynthesis), *ispADFGH* (terpenoid backbone biosynthesis) and *ribABCDH* (riboflavin biosynthesis). Loss of these genes may reflect relaxed selection on components of a pathway after inactivation of a gene within that pathway.

In addition to genes involved in metabolism, genes encoding outer membrane and secretory proteins, including *ompA, oprC, secBDF* and *tonB*, were lost independently in multiple lineages, possibly affecting communication between endosymbiont and host cells. Finally, parallel loss of five hypothetical proteins with unknown function emphasizes the need to better characterize these proteins and understand their contributions to the symbiosis. Maintenance of these hypothetical proteins in particular lineages suggests that they are under selective pressure specific to host lineages.

### *Blochmannia* lineages experienced multiple inversions

Although gene order is conserved among the four sequenced *Blochmannia* of *Camponotus*, comparisons including *B. obliquus* and *B. turneri* revealed inversions in eight genomic regions involving between two and 34 genes (Figs. 3 and 6). Two of these regions (Figs. 3D and 6D, Figs. 3G and 6G) show evidence of multiple separate inversion events (Table S3), suggesting that they may be inversion "hotspots." To determine which lineage likely experienced inversion events, we compared gene order in the eight regions to that of close relatives strain HS (CP006569), which is a member of a *Sodalis*-allied clade, and *Baumannia cicadellinicola* str. Hc (NC_007984). For some regions, these close relatives provide information on the probable gene order of the ancestral *Blochmannia* lineage. Based on these comparisons, we hypothesize that two inversions occurred in the lineage leading to *B. obliquus* (Figs. 3A/6A and Figs. 3B/6B), one occurred in the lineage leading to *B. turneri* (Figs. 3F/6F), one occurred in the ancestral lineage of *B. turneri* and

*Blochmannia* of *Camponotus* (Figs. 3C and 6C), and one occurred in the ancestral lineage of the four sequenced *Blochmannia* of *Camponotus* (Figs. 3E and 6E). For the remaining three regions, our comparisons were inconclusive.

## DISCUSSION

Comparative genomics of *Blochmannia* lineages spanning the origin of the ant tribe Camponotini allows us to reconstruct early events in the evolution of this symbiosis. Previous phylogenetic analyses identified a clade of secondary endosymbionts of mealybugs as the closest relatives of *Blochmannia*, suggesting *Blochmannia* may have originated from an ancestor of this group (*Wernegreen et al., 2009*). Reconstruction of the last common ancestor of the *Blochmannia* analyzed here showed most genes in the LCA (82%) have been retained in all six *Blochmannia* species sequenced to date. This is consistent with the hypothesis that the ancestor was an endosymbiont with an already reduced genome. Alternatively, *Blochmannia* may have undergone a similar trajectory as that proposed for *Blattabacterium*, which may have originated as a free-living associate that experienced substantial and rapid gene loss after acquiring an endosymbiotic lifestyle but before diverging into extant lineages (*Patino-Navarrete et al., 2013*). Sequencing the genomes of *Blochmannia* of other deeply divergent lineages, such as *Opisthopsis*, and closely related lineages outside of *Blochmannia* may provide data to distinguish these two hypotheses.

We found that gene content differences among *Blochmannia* lineages involve several key functions, including information transfer, metabolism, and cell–cell communication, which may affect the functioning of this mutualism across ant host lineages. Regarding metabolism, the biosynthesis pathways for two vitamins, riboflavin and vitamin B6, vary among the sequenced *Blochmannia* lineages. Plants and most bacteria encode a riboflavin biosynthesis pathway, but animals lack this pathway. Many insect endosymbionts, such as *Baumannia*, *Blattabacterium*, *Buchnera*, *Hamiltonella*, *Sodalis* and *Wigglesworthia*, synthesize riboflavin. Young aphids depend on the supply of riboflavin from *Buchnera* for growth and development (*Nakabachi & Ishikawa, 1999*). Our analyses showed that *B. obliquus* and *B. turneri* have both lost the ability to synthesize riboflavin, whereas *Blochmannia* of *Camponotus* have retained this pathway and can provision the ants with this vitamin. These differences in riboflavin biosynthetic functionality illustrate how the nutritional roles of *Blochmannia* in this mutualism change during co-evolution with hosts.

In other endosymbiont systems, loss of riboflavin biosynthesis is compensated by either a secondary symbiont or transfer of the genes to the host nuclear genome. In the aphid *Cinara cedri*, the primary endosymbiont *Buchnera* lacks riboflavin biosynthesis genes, but a more recently integrated *Serratia symbiotica* associate retains this pathway (*Perez-Brocal et al., 2006*; *Lamelas et al., 2011*). In the mealybug *Planococcus citri*, the symbiont *Moranella endobia*, which is nested within cells of the endosymbiont *Tremblaya princeps*, encodes two riboflavin biosynthesis genes. Two other genes are encoded by the host nuclear genome and appear to have been transferred from facultative symbionts during past colonizations (*Husnik et al., 2013*). In contrast to the above symbioses, our analysis of Illumina read datasets generated from genomic DNA of *C. obliquus* ants and *P. turneri* gasters showed no

evidence that other bacterial associates or the ant host encode intact riboflavin biosynthesis genes. Rather, the ant hosts may acquire riboflavin via their diet, or it is possible that members of the gut microbiome synthesize this vitamin for the host. Gut bacteria are not represented at high coverage in our read datasets.

As with riboflavin, insects lack a pathway for biosynthesis of vitamin B6 (*Tanaka, Tateno & Gojobori, 2005*) and thus rely on their diet or symbionts to supply this essential cofactor. Five of the six sequenced *Blochmannia* encode the 'DXP dependent' pathway for vitamin B6 biosynthesis characteristic of *E. coli* and other gamma-proteobacteria (*Mukherjee et al., 2011*). The *B. obliquus* lineage has lost all but two of the seven genes in this pathway. In the other *Blochmannia* lineages, these genes are scattered along the genome rather than adjacent to each other, suggesting loss due to relaxed selective pressure on the pathway instead of a large deletion event affecting multiple genes. This explanation is supported by retention of *dxs* and *serC* in *B. obliquus*. These genes are also involved in amino acid metabolism and terpenoid backbone biosynthesis pathways, respectively, which are conserved in *Blochmannia*. The two genes are likely under selective pressure due to their roles in these other metabolic pathways, which prevented their loss in the *B. obliquus* lineage.

In addition to metabolism, *Blochmannia* lineages show differences in replication and repair genes. Specifically, the chromosomal replication initiation protein *dnaA* was lost in all but one of the sequenced *Blochmannia* lineages. Some obligate bacterial endosymbionts of insects, such as *Buchnera* species, have retained *dnaA* despite severe genome reduction. By contrast, loss of *dnaA* has occurred in a few insect endosymbionts, including *Baumannia, Blattabacterium, Carsonella, Sulcia* and *Wigglesworthia* (*Akman et al., 2002*; *Nakabachi et al., 2006*; *Wu et al., 2006*; *Lopez-Sanchez et al., 2009*). DnaA is also missing from a bacterial endosymbiont of protists found in termite guts (*Hongoh et al., 2008*). Previously, it was hypothesized that loss of *dnaA* was necessary for establishing a stable symbiosis between insects and bacterial endosymbionts located in the cytosol, because it may enable direct control of symbiont DNA replication by the host (*Gil et al., 2003*). The presence of intact *dnaA* in *B. turneri* and a *dnaA* pseudogene in *B. obliquus* demonstrates that loss of *dnaA* was not required for establishment of a stable symbiosis between *Blochmannia* and camponotines. However, the precise function of DnaA in *B. turneri* and the mechanisms for controlling initiation of chromosome replication in different *Blochmannia* lineages are unclear. In addition, it remains untested whether divergent *Blochmannia* lineages live in the cytosol like *B. floridanus* or, alternatively, occupy host-derived vacuoles.

The DNA repair genes *mutM* and *mutY* are also differentially distributed in the *Blochmannia* lineages, with either one or the other retained. Loss of *mutY* appears to be more common in intracellular bacteria (*Garcia-Gonzalez, Rivera-Rivera & Massey, 2012*). Overexpression of *mutM* can "rescue" inactivation of *mutY* in *E. coli* (*Michaels et al., 1992*), which implies that retention of *mutM* may be favored more strongly by selection. However, some bacteria, including obligate intracellular *Rickettsia* species, lack both *mutM* and *mutY* (*Garcia-Gonzalez, Rivera-Rivera & Massey, 2012*). Additional sequencing of *Blochmannia*

from diverse ant hosts may reveal the evolutionary trajectories leading to differential loss of *mutM* and *mutY* in *Blochmannia*.

Obligate bacterial endosymbionts of insects are characterized by a high degree of genome stability, with strictly conserved gene order observed among sequenced representatives of some endosymbiont groups, such as *Carsonella* and *Sulcia*. Comparative genomics of other endosymbiont genera, including *Blattabacterium* and *Buchnera*, have identified typically three or fewer inversions. Recently, an exception to this extreme conservation of genome architecture was described in *Portiera*, an endosymbiont of whiteflies (*Sloan & Moran, 2013*). Analysis of *Portiera* genomes from divergent whitefly host genera predicted at least 17 inversion events, with most occurring in one lineage that also had a high prevalence of tandem repeats.

Previously, the *Blochmannia* genome dataset sampled only species from *Camponotus* hosts, and these four sequenced species shared strictly conserved gene order. By sequencing *Blochmannia* from ant hosts on divergent branches of the tribe Camponotini, our analysis revealed that inversions have occurred throughout the evolution of this endosymbiont group. Comparisons with close relatives suggest that inversions are not limited to a particular *Blochmannia* lineage, but rather may have occurred along all four major branches of the phylogeny. We identified gene losses in some of the regions involved in inversion events; it is possible that changes in mutational pressure arising from strand switch contributed to degradation and eventual loss of these genes (*Williams & Wernegreen, 2012*).

By sequencing genomes from *Blochmannia* of divergent ant host lineages, we expanded the available *Blochmannia* genome dataset beyond *Camponotus* hosts and reconstructed evolutionary trajectories of *Blochmannia* that likely span the origin of the tribe Camponotini. Analysis of deep branches in symbiont groups addresses questions surrounding the origin of symbioses and the mechanisms involved in establishment of stable associations. Although divergent *Blochmannia* genomes share much of their gene content, differential gene losses across key functional categories are likely to impact the host-bacterial partnership. It remains challenging to distinguish if different losses reflect selective fine-tuning across distinct ant hosts, stochastic gene deletions or a combination of the two. However, our results, particularly the numerous instances of parallel gene loss, hint that the strength or efficacy of selection to maintain gene functions has varied across ant host lineages and contributed to observed genome variation. These findings contribute to a broader understanding of processes shaping genome reduction in insect endosymbionts and potentially in other bacteria, including intracellular pathogens.

## ACKNOWLEDGEMENTS

The authors gratefully acknowledge Simon K.A. Robson for providing specimens of *P. turneri* and Benoit Guénard for providing *C. obliquus*. We thank Yongliang Fan for preparing genomic DNA and Lisa Bukovnik, Olivier Fedrigo and Fangfei Ye for Illumina sequencing through the core facility in the Center for Genomic and Computational Biology (GCB) at Duke University. We are grateful to Julie Yi for assistance with genome

gap closing and analysis of *tolA*. We thank Michelle G. Giglio at the Institute for Genome Sciences in the University of Maryland School of Medicine for assistance with the MANATEE annotation framework. We thank Philip S. Ward for helpful discussion and for archiving voucher specimens at the Bohart Museum of Entomology, University of California, Davis (UCDC).

### Funding

This work was supported by grants to JJW from the National Institutes of Health (R01GM062626) and the National Science Foundation (MCB-1103113). The funders had no role in study design, data collection and analysis, decision to publish, or preparation of the manuscript.

### Grant Disclosures

The following grant information was disclosed by the authors:
National Institutes of Health: R01GM062626.
National Science Foundation: MCB-1103113.

### Competing Interests

The authors declare there are no competing interests.

### Author Contributions

- Laura E. Williams performed the experiments, analyzed the data, wrote the paper, prepared figures and/or tables, reviewed drafts of the paper.
- Jennifer J. Wernegreen conceived and designed the experiments, contributed reagents/materials/analysis tools, reviewed drafts of the paper.

### DNA Deposition

The following information was supplied regarding the deposition of DNA sequences:
GenBank CP010048, CP010049.
SRA SRP050161, SRP050154.

### Supplemental Information

Supplemental information for this article can be found online at http://dx.doi.org/10.7717/peerj.881#supplemental-information.

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
