# Peer review of "Genome evolution in an ancient bacteria-ant symbiosis: parallel gene loss among Blochmannia spanning the origin of the ant tribe Camponotini"

_PeerJ, doi:10.7717/peerj.881_

## Round 0.1 · original submission · Minor Revisions

There are only some minor comments that were kindly provided by the two reviewers. I also agree with reviewer 2 on the comment regarding referring to genome reduction in marine free-living bacteria; what is the connection?

Reviewer 1 ·

Basic reporting

No comments

Experimental design

No comments

Validity of the findings

No comments

Additional comments

L20: Which nitrogen recycling pathway is being referred to?
L25: please change 'high' to 'elevated'
L29-31: please provide a reference(s) for "Gene loss is likely...vertical transmission of symbionts".
L52-53: Please state what characters (genes, intergenic regions, etc.) were used to generate the "phylogenetics evidence."
L100-101: Please state what was to be learned from the VarScan analysis.
L165-167: Does a maximum likelihood analysis recover the same phylogenetic tree as that generated by MrBayes?
L239-241: If the reads comprising the B. turneri contig and the those for the genome are reassembled, are they integrated in to the genome or do they continue to form a contig exclusive of the B. turneri genome?
L328: Quality info for these reads would clarify if these are in fact low-quality reads.
L388: Please indicate what alternative pathway is referenced in Williams and Wernegreen.
L481-483: Seems like FISH microscopy of the C. obliquus and P. turneri tissues could easily address this question.
Figure 3: please include the genome sizes or size ranges in the plots.
Figure 4: Could B6 production start with 4-phospho-D-erythronate instead of D-erythrose 4-phosphate?

Reviewer 2 ·

Basic reporting

Well written article which highlights warly events in Blochmannia genome evolution. Blochmannia genomes from two divergent host lineages (in the genera Polyrhacis and Colobopsis referred to as B. turneri and B. obliquus; respectively) were compared with 4 other publically available Blochmannia genomes.

1. Abstract, I understand the link of these findings to other intracellular bacteria, but why mention free-living marine bacteria (in Abstract and Discussion)?
2. Line 27; Provide a citation(s) for few recombination events in endosymbiont lineages.
3. Lines 130-131; Provide a citation for frameshifts may be corrected by polymerase slippage.

Experimental design

1. Lines 143-144; Was there a minimum amount of nonsense mutations, frameshifts and/or gaps necessary for the pseudogene designation?

Validity of the findings

Use of “convergent” evolution/loss in the title and through out the manuscript may be a stretch, as these processes occur in organisms not closely related/monophyletic where organisms independently evolve similar traits. Although differential gene loss between Blochmannia symbionts is certainly very interesting, convergence may not be a suitable term.

---

## Round 0.2 · accepted · Accept

The reviewers’ comments have been effectively addressed in the revision.

Reviewer 1 ·

Basic reporting

No comments.

Experimental design

No comments.

Validity of the findings

No comments.

Additional comments

No comments.